# Long-Term Effects of Biliverdin Reductase a Deficiency in *Ugt1*^−/−^ Mice: Impact on Redox Status and Metabolism

**DOI:** 10.3390/antiox10122029

**Published:** 2021-12-20

**Authors:** Giulia Bortolussi, Xiaoxia Shi, Lysbeth ten Bloemendaal, Bhaswati Banerjee, Dirk R. De Waart, Gabriele Baj, Weiyu Chen, Ronald P. Oude Elferink, Ulrich Beuers, Coen C. Paulusma, Roland Stocker, Andrés F. Muro, Piter J. Bosma

**Affiliations:** 1International Centre for Genetic Engineering and Biotechnology, 34149 Trieste, Italy; bortolussi@icgeb.org (G.B.); Bhaswati.Banerjee@icgeb.org (B.B.); 2Tytgat Institute for Liver and Intestinal Research, Amsterdam Gastroenterology and Metabolism, Amsterdam University Medical Centers, Location AMC, University of Amsterdam, 1105 AZ Amsterdam, The Netherlands; x.shi@amsterdamumc.nl (X.S.); l.tenbloemendaal@amsterdamumc.nl (L.t.B.); d.r.dewaart@amsterdamumc.nl (D.R.D.W.); r.p.oude-elferink@amsterdamumc.nl (R.P.O.E.); u.h.beuers@amsterdamumc.nl (U.B.); c.c.paulusma@amsterdamumc.nl (C.C.P.); 3Key Laboratory of Protein Modification and Disease, School of Bioengineering, Dalian University of Technology, Dalian 116024, China; 4Light Microscopy Imaging Center, Department of Life Sciences, University of Trieste, 34127 Trieste, Italy; gbaj@units.it; 5Heart Research Institute, Sydney, NSW 2042, Australia; Weiyu.Chen@hri.org.au (W.C.); Roland.Stocker@hri.org.au (R.S.)

**Keywords:** Crigler-Najjar, Gilbert, fetus, aging, Prdx2, triglycerides, iron accumulation

## Abstract

Accumulation of neurotoxic bilirubin due to a transient neonatal or persistent inherited deficiency of bilirubin glucuronidation activity can cause irreversible brain damage and death. Strategies to inhibit bilirubin production and prevent neurotoxicity in neonatal and adult settings seem promising. We evaluated the impact of *Bvra* deficiency in neonatal and aged mice, in a background of unconjugated hyperbilirubinemia, by abolishing bilirubin production. We also investigated the disposal of biliverdin during fetal development. In *Ugt1*^−/−^ mice, *Bvra* deficiency appeared sufficient to prevent lethality and to normalize bilirubin level in adults. Although biliverdin accumulated in *Bvra*-deficient fetuses, both *Bvra*^−/−^ and *Bvra*^−/−^*Ugt1*^−/−^ pups were healthy and reached adulthood having normal liver, brain, and spleen histology, albeit with increased iron levels in the latter. During aging, both *Bvra*^−/−^ and *Bvra*^−/−^*Ugt1*^−/−^ mice presented normal levels of relevant hematological and metabolic parameters. Interestingly, the oxidative status in erythrocytes from 9-months-old *Bvra*^−/−^ and *Bvra*^−/−^*Ugt1*^−/−^ mice was significantly reduced. In addition, triglycerides levels in these 9-months-old *Bvra*^−/−^ mice were significantly higher than WT controls, while *Bvra*^−/−^*Ugt1*^−/−^ tested normal. The normal parameters observed in *Bvra*^−/−^*Ugt1*^−/−^ mice fed chow diet indicate that *Bvra* inhibition to treat unconjugated hyperbilirubinemia seems safe and effective.

## 1. Introduction

The first step in the catabolism of heme is the opening of the heme ring catalyzed by heme-oxygenase (HO). In vertebrates, heme-oxygenases have selectivity for the α ring, resulting in the generation of virtually only biliverdin IXα and a small amount of biliverdin IXβ [1,2,3,4]. Both forms are non-toxic green metabolites that can be disposed via bile or urine without additional modifications [3]. In mammals, both biliverdin forms are rapidly reduced by biliverdin reductase. The ubiquitously expressed biliverdin IXα reductase (BVRA; UniProtKB: P53004; EC:1.3.1.24) converts biliverdin IXα into bilirubin IXα, a neuro-toxic and poorly water-soluble metabolite [5]. The generation of bilirubin Ixα makes heme metabolism more complex [6,7,8,9] since it needs glucuronidation by UDP-glucuronosyl transferase 1A1 (UGT1A1; UniProtKB:P22309; EC:2.4.1.17) for efficient disposal. Early after birth, hepatic glucuronidation is often insufficient to conjugate the large amount of bilirubin generated by the catabolism of fetal heme since the transcription of the *UGT1A1* gene is not yet completely activated in most neonates. This results in the accumulation of unconjugated bilirubin (UCB) in plasma and tissues, clinically observed as mild neonatal jaundice. In the large majority of cases, this condition is transient and is usually considered benign and possibly beneficial in view of the anti-oxidant properties of bilirubin [10]. However, in some babies, uncontrolled unconjugated hyperbilirubinemia does develop due to concurrent genetic and/or non-genetic causes, with the excess of UCB accumulating in specific brain regions resulting in acute bilirubin encephalopathy (kernicterus) and, eventually, death by kernicterus [11].

In high-income countries, this pathological condition is usually recognized in time and irreversible bilirubin-induced brain damage has become rare, with an incidence in the range from about 0.4 to 2.7 per 100,000 births [12,13], while the incidence in pre-term babies is significantly increased [14]. However, the situation is completely different in low- and middle-income countries [15]. For example, a recent survey in Nigeria identified acute bilirubin encephalopathy in 15.3% of hospital admissions for jaundice [16].

Affected babies are treated with intensive phototherapy (PT), combined with exchange transfusion in the most severe cases [17]. However, in low- and medium-income countries, the lack of both PT units and specialized centers to efficiently perform exchange transfusions, and delays in their implementation, contribute to the high incidence of permanent brain damage and death [15,18].

Inhibition of biliverdin reductase could be a promising and safe possibility. Individuals with mutations in the *BVRA* gene appear healthy and do not present biliverdin accumulation unless associated with other liver pathologies, and biliverdin exerts direct and *Bvra*-independent antioxidant effects [19,20,21]. Moreover, non-placental animals such as birds lack the *Bvra* gene [3,22]. Similarly, *Bvra*^−/−^ mice develop normally and reproduce at the expected Mendelian frequency [23].

Thus, the possibility to treat unconjugated hyperbilirubinemia by blocking biliverdin reductase activity and thereby bilirubin production seems an obvious strategy that has not been tested yet.

In the present work, we generated mice deficient for both *Bvra* and Ugt1 showing that the absence of *Bvra* activity in a context of severe unconjugated hyperbilirubinemia prevents bilirubin-induced neonatal lethality and brain abnormalities. Moreover, we explored the effect of biliverdin during the gestational period in *Bvra*^−/−^ embryos and its disposal. Finally, since the treatment of CNS patients will require lifelong BVRA inhibition, we investigated the safety of long-term *Bvra* deficiency by analyzing metabolism and redox status in single and double knock-out mice over time.

## 2. Materials and Methods

### 2.1. Animals

Mice were housed and handled according to institutional guidelines. The Ugt1 deficient mice reported previously (C57Bl/6 genetic background) were crossed with the *Bvra* deficient mice (also C57Bl/6 genetic background) to generate *Bvra*^−/−^*Ugt1*^−/−^ mice [23,24]. Animals were kept in a temperature-controlled environment with a 12–12 h light-dark cycle. They received a standard chow diet and water ad libitum. Genotyping of the animals was performed as previously described [23,24]. The number of animals used in each experiment is detailed in Appendix A.

### 2.2. Phototherapy Treatment

Newborn pups were exposed to blue fluorescent light (λ = 450 nm; 20 μW/cm^2^/nm; Philips TL 20W/52 lamps; Philips, Amsterdam, The Netherlands) for 12 h/d (synchronized with the light period of the light-dark cycle) as previously described [24,25]. The intensity of the lamps was monitored monthly with an Olympic Mark II Bili-Meter (Olympic Medical, Port Angeles, WA, USA).

### 2.3. Preparation of mRNA Extraction and qPCR Analysis

Isolation of tissues was performed as described previously [24]. Total RNA was isolated using Tri-reagent and cDNA was generated from 2 µg RNA using RevertAid Reverse Transcriptase (Thermo Fisher, Bleiswijk, The Netherlands). qRT-PCR was performed on a Bio-Rad CFX96 using the SensiFAST SYBR No-ROX Kit (Bioline, Waddinxveen, The Netherlands) and the primers mentioned in Appendix A. Data was normalized to mouse glyceraldehyde-3-phosphate dehydrogenase *(Gapdh*). Results were processed and analyzed using LinRegPCR software [26].

### 2.4. Blood Test Analysis

Plasma total bilirubin in young mice was determined as previously described [25]. Bilirubin, bilirubin glucuronides, and biliverdin in embryonic plasma and amniotic fluid, and in plasma, urine, and bile of 9-month-old animals obtained at the time of sacrifice, were quantified using HPLC analysis using an adapted protocol based on the method of Spivak and Carey as described [27]. Plasma biochemistry for liver injury markers (ALT and AST) was determined by routine biochemistry on a Roche Cobas c502/702 analyzer (Roche Diagnostics, Indianapolis, IN, USA).

For glucose determination and hematological analysis, animals were fasted for 4 h before facial puncture. Glucose in blood was determined using Accu-chek Guide Strips (Roche, Monza, Italy) immediately after exsanguination. Whole blood cell hematological analysis was performed with a BC-2800Vet Mindray Auto Hematology Analyzer (Mindray, Trezzano sul Naviglio, MI, Italy) with fresh EDTA-containing samples.

Reticulocytes determination was performed using acridine orange (Fluka, Rodano, MI, Italy) as previously described [28,29].

### 2.5. Preparation of Protein Extracts and Western Blot Analysis

Liver, spleen, and total brain were harvested at indicated time points, immediately frozen in liquid nitrogen, and stored at −80 °C. Tissues were lysed using 1× RIPA buffer (9806, Cell Signaling) supplemented with proteases (Complete Mini protease inhibitor, Roche, Almere, The Netherlands) and phosphatase inhibitors (PhosSTOP, Roche, Almere, The Netherlands). Protein concentration was determined by the Bradford method and 30 µg total protein extract was loaded, separated by 10% SDS Page gel, blotted onto nitrocellulose membrane that was blocked with 5% milk in PBST for 2 h to prevent a-specific binding. *Bvra* was detected using the anti-*Bvra* primary antibody (1:1000, ADI-OSA-450, Enzo, Pero, MI, Italy) incubated overnight in 5% BSA, PBST. Anti-actin (Sigma-Aldrich, Milano, Italy) antibody or Anti-HSP70 (ADI-SPA-185D) were used as loading controls.

### 2.6. Peroxiredoxin 2 (Prdx2) Oxidized Forms in Freshly Isolated Erythrocytes

Prdx2 redox status was determined as previously described by Chen et al. [23]. Briefly, blood was collected in EDTA-containing tubes and immediately mixed with 200 mM N-ethylmaleimide (Sigma-Aldrich, Milano, Italy) in a 1 to 1 (vol/vol) ratio and incubated at room temperature for 1 h. Samples (dilution 1:1000) were mixed to a non-reducing sample buffer (NP0007, Invitrogen, Monza, Italy) and stored at −80 °C until used. Five microliters of each sample was separated on 12% SDS-PAGE and blotted onto a nitrocellulose membrane. Membranes were blocked with 5% BSA in PSBT for 2 h and incubated overnight with anti-Prdx2 primary antibody (1:1000, Sigma-Aldrich, Milano, Italy) in 5% BSA PBST. Anti-rabbit HRP-conjugated antibody (Dako, Abcoude, The Netherlands) was used. Protein quantification was performed with Image Lab (Biorad, Laboratories B.V; Veenendaal, The Netherlands).

### 2.7. Triglyceride Quantification

Lipid extraction from the liver was based on the method described by Srivastava et al. [30]. Briefly, 50 mg of liver tissue was homogenized and lipid was extracted using methanol:chloroform (1:3, *v/v*), and dissolved in 1 mL 2% Triton X-100 (Bio-Rad Laboratories B.V; Veenendaal, The Netherlands). Triglyceride content in 5 µL of lipid extracted from 50 mg liver or 5 µL heparin plasma was determined using the triglycerides/glycerol blanked enzymatic colorimetric assay (Roche, Almere, The Netherlands) according to the instructions.

### 2.8. Iron Quantification

Extraction of non-heme iron from the spleen was based on the method described by Whittaker et al. [31]. Briefly, 100 mg spleen tissue was homogenized in 15 mL water. To 3 mL of the homogenate, 10 mL acid reagent (6 M HCl and 1.2 M tri-chloroacetic acid; 1:1, *v/v*) was added, mixed, and incubated for 20 h at 65 °C. After cooling to room temperature, the mixture was centrifuged (1500× *g*, 20 min) and iron in the supernatant was quantified by pipetting 20 µL into a well of a 96-well plate and adding 180 µL of freshly prepared bathophenanthroline color reagent prepared by dissolving 62.5 mg bathophenanthroline disulfonic acid and 0.25 mL thioglycolic acid in 25 mL water. The final color reagent was a solution of the bathophenanthroline color reagent, sodium acetate (4.5 M), and water (1:20:20, *v/v/v*). Upon mixing, the plate was incubated for 10 min at RT before the absorbance at 535 nm was measured in a Clariostar analyzer (BMG Labtech, Offenburg, Germany).

### 2.9. Histology

The liver and spleen were fixed in 4% formaldehyde PBS solution and embedded in paraffin. Sections (4.5 µm) were stained with Hematoxylin (Sigma, 51275, Zwijndrecht, The Netherlands) and Eosin (Sigma, E4382, Zwijndrecht, The Netherlands) (H&E) or Sirius red [32]. Iron deposition in the liver and spleen was visualized using Prussian blue staining [33]. The sections were incubated in a mix of potassium ferrocyanide (Brocades, Meppel, The Netherlands) and hydrochloric acid (1 part 2% hydrochloric acid and 1 part 2% potassium ferrocyanide) for 45 min and counterstained with nuclear fast red (Sigma) for 5 min.

Immunofluorescence analysis of brain samples was performed as previously described [24]. The study was performed in a double-blind fashion: the genotype of the animals was unknown to the operator, and a different investigator analyzed the data. Measurements were averaged for each animal.

For anti-calbindin staining specimens (14 µm) were incubated with anti-calbindin antibody (1:200, Synaptic Systems, Goettingen, Germany) for 2 h at RT in blocking solution. After washings specimens were incubated with a secondary antibody (Alexa Fluor 488; Invitrogen Carlsbad, CA, USA) for 2 h at RT. Nuclei were visualized with Hoechst (10 μg/mL, Invitrogen, Bleiswijk, The Netherlands) after secondary antibody solution.

Sections were mounted in Mowiol 4-88 (Sigma-Aldrich, Zwijndrecht, The Netherlands). Images were acquired on a Nikon Eclipse E-800 epi-fluorescent microscope with a charge-coupled device camera (DMX 1200 F; Nikon Amstelveen, Amsterdam, The Netherlands). Digital images were collected using ACT-1 (Nikon) software (Amsterdam, The Netherlands).

Analysis of the layer thickness by measuring the layer depth (μm) was performed as previously described [24]; while PC density analyses were performed as previously described [24,25].

### 2.10. Statistics

Data are presented as the mean values ± SD and were tested for normal distribution (Kolmogorov-Smirnov test) before analyzing for significance using one-way analysis of variance (ANOVA) for the comparison of 3 or more groups. For statistical analysis, we used GraphPad Prism 8 software (GraphPad Software Inc., San Diego, CA, USA). * *p* < 0.05, ** *p* < 0.01, *** *p* < 0.001 were considered significant.

### 2.11. Study Approval/Ethic Statement

Experimental procedures were approved by the International Centre for Genetic Engineering and Biotechnology (ICGEB) board and by the Italian Ministry of Health (authorization N. 523/2017-PR from the Italian Ministry of Health). All experiments involving animals were conducted in full respect of the ARRIVE principles.

## 3. Results

### 3.1. Deleting Bvra Prevents Lethal UCB Accumulation in Neonatal Ugt1^−/−^ Mice

We first investigated if the sole absence of the *Bvra* gene in genetically induced unconjugated hyperbilirubinemic mice would be sufficient to block bilirubin production in *Ugt1*^−/−^ mice and prevent neonatal UCB-induced lethality [24]. Thus, we generated *Bvra*^−/−^*Ugt1*^−/−^ animals by double heterozygous crossing (*Bvra*^+/−^*Ugt1*^+/−^). Initially, all pups were temporarily treated with phototherapy (PT) until postnatal day 20 (P20) (Appendix A), an experimental setup shown to prevent neonatal death due to bilirubin neurotoxicity, as previously reported for *Ugt1*^−/−^ mice [24,25]. All genotypes survived up to P30 without any sign of toxicity (Figure 1A,B and Appendix A). At P30, in *Bvra*^−/−^ mice total plasma bilirubin levels (TB, determined at sacrifice) were below 0.1 mg/dL (the lower limit of detection). As expected, TB levels in the *Ugt1*^−/−^ group were very high, in the range of 10 mg/dL, confirming previous results [24,25]. Importantly, at P30 *Bvra*^−/−^*Ugt1*^−/−^ animals had plasma bilirubin levels of 0.1 mg/dL, similar to those in WT control mice (<0.1 mg/dL) (Figure 1B).

To investigate if *Bvra* deficiency in a hyperbilirubinemic background is protective, a second group of newborn mice was not treated with PT (Appendix A). This very challenging condition resulted in the death of all *Ugt1*^−/−^ mice within the first week of life due to hyperbilirubinemia-related neurotoxicity confirming previous findings [24]. Strikingly, all *Bvra*^−/−^*Ugt1*^−/−^ mice survived (Figure 1C).

To determine possible neurological damage, mice were sacrificed at postnatal day 4 (P4), and brain and blood were collected. Plasma bilirubin level at P4 in the *Bvra*^−/−^*Ugt1*^−/−^ animals were 0.2 mg/dL, comparable to that in WT mice (0.1 mg/dL), and, as expected, very high in the *Ugt1*^−/−^ animals and not detectable in the *Bvra*^−/−^ mice (Figure 1D). Brain histology at P4 showed no abnormalities in the *Bvra*^−/−^ and *Bvra*^−/−^*Ugt1*^−/−^ groups (Figure 1E,F), while the *Ugt1*^−/−^ animals presented important cerebellar abnormalities, with a significant decrease in the depth of the external germinal layer and in the number of Purkinje cells, in line with previously reported data of P5 pups [24].

Overall, these results indicated that preventing *Bvra* enzyme activity (in our case by a null mutation in the *Bvra* gene) is sufficient to block the production of the hydrophobic and neurotoxic bilirubin IXa.

### 3.2. Biliverdin Accumulation in Bvra Deficient Embryos Does Not Affect Viability

The percentage of *Bvra*^−/−^ pups generated by cross-breeding of Brva^−/−^ and *Ugt1*^−/−^ showed no significant difference with the expected frequency, based on Mendelian inheritance (Appendix A). Thus, the absence of biliverdin reductase A during embryonic development of *Bvra*^−/−^*Ugt1*^−/−^ does not result in toxicity that would affect embryonal viability, in line with the original report on *Bvra*^−/−^ mice [23].

In addition, this implies that the hypothesis that the *Bvra* occurrence in mammals is required for life birth should be revised.

However, the presence of *Bvra* in the placenta suggests a role related to heme turn-over in the gestational phase [3,34]. In order to shed light to this issue, the effect of *Bvra* absence in the placenta and embryos was investigated. Breeding of *Bvra*^+/−^ or *Bvra*^−/−^ females with *Bvra*^−/−^ males was set up and embryos and placentas were analyzed at embryonic day 18.5 (E18.5). All *Bvra*^−/−^ embryos, and especially the placenta, had a readily recognizable greenish color irrespective of the presence or absence of *Bvra* in the mother (Figure 2A,B). Analysis of the amniotic fluid and plasma demonstrated the presence of biliverdin only in *Bvra*^−/−^ embryos, but not in wild-type embryos (Figure 2C,D). Bilirubin (~2 µM) was present in the plasma but not in the amniotic fluid of wild-type embryos. The total amount of heme metabolites in *Bvra*-deficient embryos (10 µM biliverdin in the amniotic fluid and 4 µM in plasma) was significantly higher compared to that of WT embryos (2 µM bilirubin in plasma only), indicating that the conversion of biliverdin to bilirubin indeed results in a more efficient clearance from the embryo into the maternal circulation.

These results demonstrate that, in the absence of *Bvra*, biliverdin IXα produced in the embryo cannot be reduced to bilirubin IXα and cannot cross the placental barrier as efficiently as bilirubin. The impaired transfer to the maternal circulation results in biliverdin accumulation in the embryonic sac and the amniotic fluid. Finally, the presence of a significant amount in the amniotic fluid demonstrates that the embryonic kidney can eliminate biliverdin.

### 3.3. Biliverdin and Bilirubin Accumulation in Bvra Deficient Mice during Aging

The evidence that *Bvra* activity is not essential for live births leaves room for other explanations regarding the occurrence of the very efficient reduction in biliverdin IXa seen in mammals. One view is that bilirubin IXa is not just a waste product, but functions as an anti-oxidant and as such is protective towards oxidative stress [10,35]. To investigate whether the absence of bilirubin would result in long-term tissue damage due to increased reactive oxygen species (ROS) and oxidative stress, the effects of *Bvra* gene deficiency were investigated in animals bearing the WT or mutant *Ugt1* gene. Only *Ugt1*^−/−^ mice were treated with PT up to 20 days after birth to prevent lethal UCB brain damage [24]. Irrespective of their genotype, all mice survived without showing any obvious abnormality at the time of sacrifice (9 months of age).

Bilirubin accumulation was only seen in *Ugt1*^−/−^ mice, resulting in high plasma levels and just a small amount in bile (Figure 3A,B). As expected, bilirubin glucuronides were detected only in the bile of WT animals.

In all four animal groups (WT, *Bvra*^−/−^, *Ugt1*^−/−^ and *Bvra*^−/−^*Ugt1*^−/−^), biliverdin levels in plasma were low, in the nM range, indicating that, in contrast to bilirubin IXa, this water-soluble metabolite does not accumulate to high levels (Figure 3A). The presence of significant levels of biliverdin in bile and urine in all *Bvra*-deficient mice demonstrates that the liver and kidneys efficiently secrete it (Figure 3B,C). In fact, the gallbladder of these animals was green and the urine had a greenish color, indicating the presence of biliverdin (Figure 3D).

*Bvra* protein was absent in the brain, spleen, and liver of neonate P4 and adult P60 *Bvra*^−/−^ mice, while *Ugt1*^−/−^ mice had comparable expression levels to WT (Appendix A). The expression of *Bvra* in the brain and spleen of adult mice was higher than in neonates, while in the adult liver it was lower (Appendix A).

To evaluate potential long-term adverse effects of *Bvra* inhibition, liver enzymes in serum and liver histology of middle-aged (9-month-old animals) mice were compared to control mice (Appendix A). We observed no statistically significant increase in AST and ALT in any of the mutant groups. Interestingly, the ALT levels were significantly lower in both Ugt1 deficient mouse lines compared to WT mice, (Appendix A). All mouse strains presented normal liver architecture, morphology, and weight, with the absence of fibrosis, as determined by H&E and Sirius red staining (Appendix A). Overall, these data do not reveal any overt long-term liver damage or toxicity due to *Bvra* deficiency.

### 3.4. Aging-Related Systemic Oxidative Stress in Bvra Deficient Mice

Whole blood hematological analysis in 3-months old mice revealed statistically significant differences in the total white blood cell counts compared to WT, as well as in the counts of their three major white cell types (lymphocytes, monocytes, and granulocytes), although the absolute differences were minor and absolute counts were all in the normal range (Appendix A). Importantly, all the observed differences disappeared over time and were absent in 9-month-old animals (Appendix A). Similarly, minor but statistically significant differences were observed in red blood cell counts, RBC-related parameters, and platelets counts, which were all within normal values. (Appendix A).

The percentage of reticulocytes in the blood is a well-established parameter for the generation of erythrocytes. We reported a significant decrease in both *Bvra*^−/−^ and *Bvra*^−/−^*Ugt1*^−/−^ mice at adult age (3 months old), both in males and females (Figure 4A) as determined by acridine orange staining. These data indicate that the production of erythrocytes in *Bvra* deficient mice is reduced. The absence of a major difference in red blood cells counts (Figure 4B) suggests a decreased turnover and longer lifespan of erythrocytes in young *Bvra*^−/−^ mice, resulting in a longer presence in the circulation and, thus, a potentially longer exposure of these cells to putative systemic oxidative stress.

This putative prolonged exposure to systemic oxidative stress coincided with the increased levels of oxidized peroxiredoxin 2 (Prdx2), one of the major proteins of red blood cells and an established marker for oxidative stress in erythrocytes, seen in both young *Bvra*-deficient strains (Figure 4C and Appendix A), and as reported previously [23]. Unexpectedly, in 9-month-old mice the ratio of oxidized vs. reduced Prdx2 in both *Bvra*^−/−^ and *Bvra^−/−^Ugt1^−/−^* mouse strains was significantly decreased compared to WT and Ugt1 deficient mice in both males and females (Figure 4D,E and Appendix A). This seems to indicate compensation of systemic oxidative stress during aging in *Bvra* deficiency. The comparable percentages of all other cells in the blood of these older mice also did not provide an indication of a potential mechanism (Figure 4F and Appendix A).

Since the heme breakdown generates biliverdin and iron predominantly in the reticuloendothelial system, we next investigated the long-term effects of *Bvra* deletion in the spleen. We examined non-heme iron disposition in tissues as hemosiderin, which can be identified histologically by Perl’s Prussian blue. Interestingly, Prussian blue staining revealed significant iron deposition in the spleen of *Bvra^−/−^* and *Bvra*^−/−^*Ugt1^−/−^* mice (Figure 5A), but not in the liver (Appendix A). Hemosiderin deposits mostly are asymptomatic, but a more than 3-fold increase in iron accumulation in the spleen has been reported to cause inflammation and to lead to damage of splenocytes [36]. The 2-fold higher non-heme iron content in the spleen of 9-month-old mice (Figure 5B) did not result in increased expression of the inflammation markers interleukin-6 (*Il6*, *IL-6*) and tumor necrosis factor (*Tnf, TNF-**α*, Appendix A) nor an increase in spleen weight (Figure 5C). The liver mRNA level of hepcidin, a protein that inhibits intestinal iron uptake and iron export from the spleen, did not differ significantly, suggesting the absence of a major change in iron metabolism (Appendix A) [37]. Still, the increased iron disposition suggests long-term *Bvra* deficiency affects iron metabolism.

In addition to its enzymatic function, *Bvra* has been proposed to have a role in regulating cellular signaling [38]. In this respect, its role in the regulation of heme oxygenase 1 (*Hmox1,* HO-1) expression is of interest. The HO-1 gene expression in liver, spleen, and brain in 9-month-old *Bvra^−/−^* and *Bvra*^−/−^*Ugt1^−/−^* mice did not differ from that in WT mice (Figure 6A). The similar mRNA levels in these tissues render the role of *Bvra* in regulating HO-1 expression in vivo unlikely.

Gene expression analysis of other oxidative stress-related genes, such as nuclear factor, erythroid-derived 2, like 2 (*Nfe2l2, Nrf2*, a transcription factor that regulates genes which contain antioxidant response elements (ARE) in their promoters), NAD(P)H dehydrogenase, quinone 1 (*Nqo1*, a member of the NAD(P)H dehydrogenase family, and target gene of Nrf2), and glutamate-cysteine ligase, modifier subunit (*Gclm*, the first rate-limiting enzyme of glutathione synthesis) in the liver, spleen, and brain showed no major differences among the genotypes, except for an increase in Nqo1 and Gclm in the spleen of *Bvra*^−/−^*Ugt1^−/−^* mice (Figure 6B,D). Together, these results suggest that contrary to the situation in circulation, *Bvra* deficiency does not cause substantive oxidative stress in tissues under the experimental conditions studied.

### 3.5. Metabolic Effects of Bvra and/or Ugt1 Deficiency in Aged Mice

Both biliverdin and bilirubin have been reported to be involved in the activation of peroxisome proliferator-activated receptor-alpha (PPARα), a central player in β-oxidation and lipid metabolism [39]. PPARα expression is involved in the regulation of fatty acid oxidation, fatty liver progression, body weight, and blood glucose. In addition, since serum bilirubin levels have been negatively associated with hypertriglyceridemia [35], we analyzed the levels of triglycerides in plasma and liver, as well as the expression of PPARα, and key genes regulated by PPARα involved in glucose and lipid oxidation metabolisms in the liver.

Quantitative RT-PCR analysis showed that in *Ugt1*^−/−^ mice, with high serum bilirubin levels, hepatic mRNA expression levels of PPARα were moderately but significantly increased (Figure 7A) In contrast, the low bilirubin levels in *Bvra^−/−^* mice, compared to WT and double knockout mice, did not result in reduced peroxisome proliferator-activated receptor alpha (Ppara, Pparα) mRNA levels. The moderate effect of the very different bilirubin serum levels on hepatic PPARα expression in the mouse lines suggests the role of bilirubin in regulating fatty acid oxidation via PPARα at best is small.

Interestingly, a significant effect on serum triglycerides was seen. Compared to WT animals, serum triglycerides were significantly increased in *Bvra^−/−^* mice, and significantly decreased in the *Ugt1^−/−^* mice, but normal in *Bvra^−/−^Ugt1^−/−^* animals (Figure 7B).

Importantly, despite the important differences observed in plasma triglyceride values between the single KO strains and WT mice (Figure 7B), no significant differences were seen in liver triglyceride levels among the different groups (Appendix A). Furthermore, hepatic mRNA levels of PPARα-responsive genes, carnitine palmitoyl transferase 1a (*Cpt1a*), glycogen synthase 2 (*Gys2*) and fibroblast growth factor 21 (*Fgf2*1), in *Bvra^−/−^* and *Bvra^−/−^Ugt1^−/−^* mouse strains were comparable to those present in WT mice (Appendix A–D). In the *Ugt1*^−/−^ mice, Gys2 levels were significantly increased compared to wild-type mice (Appendix A), probably associated with the high bilirubin level in that model.

Several studies have shown that moderate levels of bilirubin are correlated with reduced abdominal obesity and a lower risk of metabolic syndrome [40,41,42,43]. Compared to wild-type mice, the body weight in the 9-months-old mice was significantly reduced in female but not in male *Ugt1*^−/−^ mice (Appendix A). A similar gender-specific effect was seen in the hyper-bilirubinemic *Ugt1^−/−^* rat caused by increased hepatic mitochondrial biogenesis in females [44]. Furthermore, the double knockout females also had a reduced body weight while their serum bilirubin levels were normal. Therefore, this gender-specific effect on body weight was probably not related to the levels of serum bilirubin. Instead, the role of Ugt1 in the metabolism of 17B-estradiol, resulting in increased serum levels of the female hormone in *Ugt1^−/−^* and the *Bvra^−/−^Ugt1^−/−^* mice seems a more likely explanation since this hormone was shown to increase mitochondrial complex IV activity [45].

Finally, since it has been reported that biliverdin and bilirubin may protect from glucose intolerance and diabetes [46,47,48,49], the glucose levels in serum were determined in fasted 3- and 9-months-old mice. In contrast to previous studies, the high serum bilirubin levels in *Ugt1^−/−^* mice did not result in lower blood glucose levels (Figure 7C and Appendix A). Overall, glucose levels in fasted animals tested normal for all genotypes.

## 4. Discussion

We demonstrated here that genetic inactivation of *Bvra* fully rescues *Ugt1^−/−^* mice from brain damage and neonatal death caused by the accumulation of neurotoxic unconjugated bilirubin early after birth. *Bvra^−/−^Ugt1^−/−^* mice were born safely at the expected Mendelian frequency and did not present any evident abnormality or pathology at birth or at older ages. This observation and previous findings that individuals and mice lacking BVRA activity are healthy [19,20,21,23], supports the idea that pharmacological inhibition of BVRA activity could be a valid approach to treat both the neonatal pathological unconjugated hyperbilirubinemia and the inherited severe form of Crigler-Najjar syndrome, reducing the risk of irreversible neurological damage. There is a clear medical need to develop efficient therapies to control unconjugated bilirubin production, especially to provide low- and middle-income countries with therapeutic alternatives to exchange transfusion and/or intensive phototherapy to treat acute pathological unconjugated hyperbilirubinemia, for which specialized pediatric wards and PT devices are often not available [15]. Identification of effective BVRA inhibitors, that are safe in vivo, would allow translation and testing of this strategy in a clinical phase 1 study.

The inhibition of the first step in the catabolism of heme, the oxidation of heme to biliverdin by HO-1, has been tested in animal models and subsequently in clinical studies [50,51,52]. Although effectively reducing bilirubin production, the adverse effects seen in patients and animal models blocked its clinical application [53,54,55]. Inhibition of the second step in heme catabolism, the reduction in biliverdin to bilirubin catalyzed by biliverdin reductase, has been proposed by our group [27]. In that study, we identified FDA-approved drugs able to block BVRA activity in vitro. Despite their potent inhibitory activity, in vivo testing failed to decrease bilirubin levels in an animal model. Dose-limiting toxicity of these drugs did not allow effective inhibition of biliverdin reduction activity by ubiquitously expressed BVRA.

The conversion of biliverdin IXα to bilirubin IXα is puzzling since it adds another step of complexity to heme catabolism in mammals, not seen for instance in birds, most reptiles, and other vertebrates [56,57,58]. One hypothesis is that mammals need BVRA activity for the efficient disposal of heme degradation products generated in the gestational phase [3]. The high expression level of BVRA in the placenta does support such a role and biliverdin administered to embryos appeared in maternal blood almost entirely as bilirubin [3,59]. Although this demonstrates that BVRA is involved in the efficient disposal of biliverdin from the embryo, data showing toxicity of biliverdin accumulation are lacking. In the first period of the embryonal growth, bilirubin IXβ, resulting from biliverdin IXβ reduction by BVRB, is the major heme-derived metabolite identified [4]. This water-soluble bilirubin isomer is efficiently disposed by the liver into the embryonal intestine preventing potential toxicity [60]. Biliverdin IXα, more abundantly found at later embryonal stages, is reduced by BVRA resulting in bilirubin IXα. The formation of this hydrophobic compound facilitates diffusion across the placenta preventing further accumulation of heme me-tabolites in the embryo. The presence of µM concentrations of biliverdin IXα in serum and amniotic fluid underscores the importance of *Bvra* in the efficient disposal of heme degradation products produced in the embryo across the placental barrier (Figure 2). This accumulation, however, did not affect embryonal viability (Appendix A, [23]). Whether the disposal of biliverdin, considered a non-toxic metabolite, into the amniotic fluid by the kidney is important to protect the embryo from potential toxicity or if higher biliverdin levels would be well tolerated by the embryo is not clear. Importantly, the lack of overt pathology in *Bvra^−/−^* and *Bvra*^−/−^*Ugt1^−/−^* deficient mice and humans demonstrates that the hypothesis claiming that generation of bilirubin IXα is an essential adaptation for embryonic life and birth can be revised.

After birth, the biliverdin serum levels decreased indicating that its transport into urine and bile is more efficient compared to that across the placenta (Figure 2 and Figure 3). The presence of biliverdin in bile and urine of *Bvra*-deficient adult animals (*Bvra^−/−^Ugt1^−/−^* and *Bvra^−/−^* mice) underscores the efficient disposal of this metabolite by the liver and the kidney.

Bilirubin has antioxidant properties that protect against reactive oxygen species and as such may not just be a waste product [10]. Such a function may explain the correlation seen between mildly increased serum bilirubin levels in subjects with Gilbert’s syndrome and a lower risk of heart disease [61,62,63,64]. In addition, *Bvra* deficiency in mice fed normal chow was associated with increased systemic oxidative stress [23]. These findings suggest that BVRA may have evolved for other needs, although these diseases occur after the reproductive age rendering the evolutionary pressure questionable.

A role of bilirubin’s redox activity in the brain has recently been reported, showing that *Bvra*^−/−^ animals presented irregular NMDAR activity when challenged with agonist or antagonist compounds [65]. We have not seen any abnormal behavior in 9-month-old *Bvra*-deficient mice, an observation supported by the absence of any evident neurological phenotype in patients lacking BVRA activity [19,20]. Further studies are required to elucidate the putative neurological effects of both Ugt1a1 and *Bvra* absence in the presence of more challenging conditions.

### 4.1. Bvra Deficiency, Oxidative Stress, and Iron Accumulation

It has been proposed that BVR participates in an antioxidant redox amplification cycle by which low, physiological bilirubin concentrations confer potent antioxidant protection via recycling of biliverdin from oxidized bilirubin by BVRA [9]. However, the existence and role of this antioxidant redox cycle have been questioned [66,67]. Still, several recent studies indicate that bilirubin is not just a waste product. As a lipid-soluble anti-oxidant, bilirubin can protect against oxygen-radical damage and lipid peroxidation [10,68,69]. In view of this antioxidant role of biliverdin and bilirubin, long-term inhibition of BVRA could cause damage rendering it not suitable for the life-long treatment of Crigler-Najjar patients. Nevertheless, the identification of healthy BVRA-deficient subjects and mice suggests that long-term BVRA inhibition may be safe [19,20,21]. The presence of WT levels of bilirubin in *Bvra^−/−^Ugt1^−/−^* mice is unexpected and it is not clear where it is generated. We speculate that the reducing environment of the intestine could result in the generation of some UCB that can enter the circulation via diffusion. A normal serum bilirubin level was also found in both BVRA-deficient subjects, suggesting a similar mechanism in humans [19].

The generation of *Bvra^−/−^Ugt1^−/−^* mice, resulting in major differences in bilirubin and biliverdin levels, made it feasible to study their role in both inflammation and metabolism in mice.

As previously reported by Chen et al. [23], erythrocytes from 3-month-old *Bvra*-deficient mice showed increased Prdx2 oxidation, indicating enhanced systemic oxidative stress (Figure 4). Surprisingly, this increase was transient. In 9-months-old *Bvra^−/−^Ugt1^−/−^* mice, an increased RBC Prdx2 redox state was seen compared to WT controls (Figure 4). A possible cause for the observed changes in the RBC Prdx2 redox state could be a change in the turnover of the erythrocytes over time. Effects of BVRA on erythroid differentiation have been reported previously. Overexpression of BVRA stimulated erythroid differentiation, while BVRA silencing inhibited erythroid differentiation in the human erythrocyte cell line K562 [70]. A decreased turnover, as observed in adult *Bvra*-deficient animals, would result in an increased lifespan of erythrocytes. In turn, this will also increase their exposure to oxidative stress in the circulation and, probably, Prdx2 oxidation. However, other mechanisms may be also present in 9-months-old mice since those animals showed normal hematological data, with no increase in reticulocyte counts or any other signs of anemia, such as lower RBC counts, hemoglobin, MCV, or hematocrit (Appendix A). In fact, minor but statistically significant differences in some parameters of the hematological analysis observed in 3-month-old mice, were, in most cases, not present in older animals. Importantly, for all examined parameters, the absolute values were within normal ranges. Further experiments may be required to better understand the biological significance of those minor variations.

Senescent or damaged erythrocytes are phagocytosed by splenic macrophages, and free iron and biliverdin are released from hemoglobin. Ferroportin exports the iron from the macrophages, and free iron binds to transferrin and is transported to the liver by portal circulation. Hepcidin, a small peptide expressed by the liver, regulates this iron turnover by modifying iron uptake from the intestine and regulating the level of ferroportin [71]. Hepcidin mRNA levels (*Hamp*) were similar in all four genotypes suggesting no effect of *Bvra* deficiency on the ferroportin–hepcidin axis (Appendix A). Another potential mechanism is the opening of the heme ring, the first step in heme catabolism, catalyzed by HO-1. Since *Bvra* has been reported to affect HO-1 expression [38], we investigated if reduced expression of this enzyme could explain the observed iron accumulation in the spleen of *Bvra*-deficient mice. However, no effect of *Bvra* deficiency was seen on HO-1 mRNA levels in any of these tissues, confirming the normal hepatic HO-1 expression reported previously [23]. In view of the high homology between the mouse and human, a role for *Bvra* in regulating HO-1 expression in vivo seems questionable. In vitro experiments showed that, in the absence of biliverdin reductase, the release of biliverdin from HO-1 is the rate-limiting step [72]. This role of *Bvra* in heme catabolism could be related to the accumulation of iron observed in the spleen of *Bvra^−/−^* and *Bvra^−/−^Ugt1^−/−^* mice.

### 4.2. Biliverdin and Bilirubin in Triglyceride and Glucose Metabolism

Bilirubin produced by BVRA has also been reported to function as a PPARα agonist lowering blood glucose and lipid accumulation [39]. The significant increase seen in mRNA levels of PPARα and a Pparα-responsive gene (*Gys2*) only in the liver of the hyperbilirubinemic *Ugt1*^−/−^ mice compared to WT mice, do support the role of bilirubin as a Pparα agonist (Figure 6 and Appendix A). However, we did not observe any significant difference in Pparα and Pparα-responsive genes tested in *Bvra^−/−^* mice, that do have lower bilirubin levels compared to WT mice, suggesting that the role of bilirubin as a Pparα agonist in vivo at low bilirubin levels is minor, if any.

The situation differed for plasma triglycerides. While adult *Bvra*-deficient mice presented no differences in the plasma triglyceride profile [23], they were significantly increased in older *Bvra^−/−^* animals. Importantly, triglycerides were significantly decreased in *Ugt1^−/−^* mice, while they were similar to WT in the double KO mice (Figure 7B). Triglycerides in the liver were in the normal range in all mouse groups, and there was no significant impact on growth curves of neonatal mice and body weight (Appendix A), providing additional support to the safety of long-term BVRA inhibition in a hyperbilirubinemia condition. To note, in this study, all animals received normal chow and no dietary, metabolic or inflammatory challenges were imposed.

*Bvra* can act on the insulin receptor kinase cascade and protect against glucose intolerance [73]. Liver-specific *Bvra^−/−^* was found to have reduced Pparα activity and increased plasma glucose and insulin levels, with increased hepatic steatosis [39]. In young and old animals, the fasted glucose test did not reveal any overt glucose intolerance in both *Bvra* deficient models (Figure 7C and Appendix A). The observed discrepancy suggests that insulin signaling seems less affected in global *Bvra*-deficient mice, fed a chow diet, than in liver-specific *Bvra*-deficient animals. *Bvra^−/−^* mice, fed with a high-fat diet, have comparable fasted plasma glucose and insulin, glucose and insulin tolerance, glucose uptake, as well as insulin signaling, but developed a fatty and inflamed liver as a result of enhanced oxidative stress reflected in increased non-enzymatic lipid peroxidation [74]. Thus, in the context of *Bvra^−/−^Ugt1^−/−^* and *Ugt1^−/−^* mice, further studies with more specific glucose and diet challenges may be needed to reach more definitive conclusions.

## 5. Conclusions

Overall, the accumulation of biliverdin during the gestational period confirms the role of *Bvra* in its efficient disposal from the embryo. The normal percentage of *Bvra*-deficient pups born, and the normal growth of *Bvra*-deficient mice indicate that the conversion of biliverdin to bilirubin is not an essential adaptation for embryonic life and birth in mammals. Nevertheless, the complete rescue of *Ugt1^−/−^* mice from lethal hyperbilirubinemia by genetically blocking bilirubin production in the *Bvra^−/−^Ugt1^−/−^* mice, together with the absence of overt pathologies in middle-aged animals, provide strong experimental support to the postulate that inhibition of BVRA is a feasible option to treat severe unconjugated hyperbilirubinemia in both hyperbilirubinemic neonates and Crigler-Najjar syndrome patients. Thus, the identification of a *Bvra* inhibitor that could be used in vivo renders additional in vitro screening of drug libraries or modification of inhibitors is compelling.

The transient mild systemic oxidative stress in adult *Bvra*-deficient mice suggests long-term inhibition of this enzyme to reduce serum bilirubin levels seems feasible albeit that the low bilirubin levels compared to man may imply that its role as an antioxidant may be less important in mice. The splenic iron accumulation observed in 9-month-old *Bvra* and *Bvra^−/−^Ugt1^−/−^* mice did not result in increased inflammation; however, since the mechanism could not be clarified, monitoring serum iron content and ferritin should be considered during long-term *Bvra* inhibition.

Finally, the compensatory effect observed in TGs levels of *Bvra^−/−^Ugt1^−/−^* compared to single knockouts are particularly intriguing and suggest an interaction between *Bvra* and *Ugt1* genes in lipid metabolism rendering additional studies needed to clarify the mechanism.

## Figures and Tables

**Figure 1 antioxidants-10-02029-f001:**
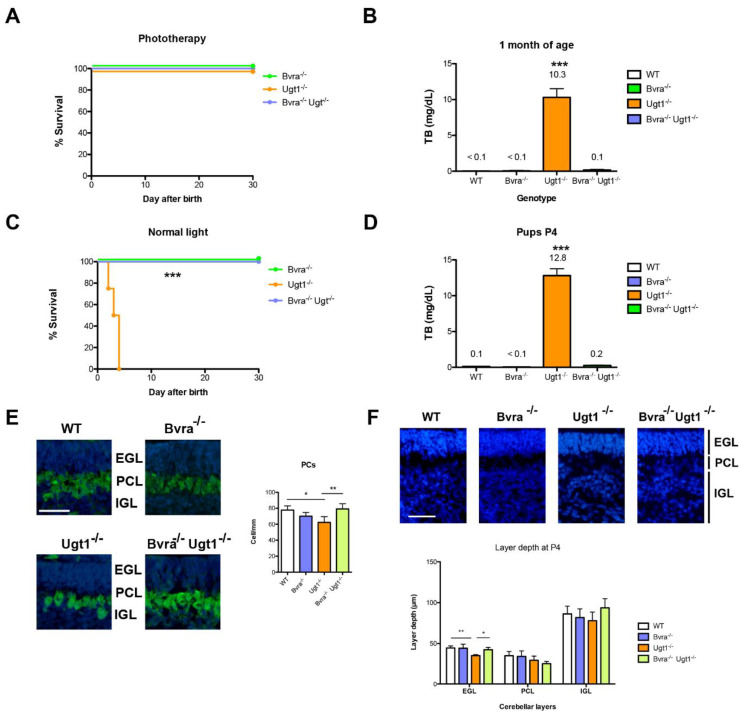
**Deleting *Bvra* prevents lethal bilirubin accumulation in *Ugt1^−/−^* mice.** (**A**); Kaplan–Meier survival curve with phototherapy treatment. Log-rank Mantel–Cox test, ns; (**B**) At P30 blood was collected and plasma total bilirubin levels (TB) were determined. One-way ANOVA, ***, *p* < 0.0001. Bonferroni post-test, *Ugt1*^−/−^ vs. all other genotypes ***, *p* < 0.0001; comparison between other genotypes, ns. (**C**) Kaplan–Meier survival curve (without PT treatment). Log-rank Mantel–Cox test, ***, *p* < 0.0001. (**D**) Total plasma bilirubin levels at P4 of the animals shown in Panel C. One-way ANOVA, ***, *p* < 0.0001. Bonferroni post-test, *Ugt1*^−/−^ vs all other genotypes ***, *p* < 0.0001; comparison between other genotypes, ns; (**E**) Cerebellar layer thickness was determined in P4 animals using Hoechst-stained brain sections. Representative images are shown in the upper panel. Scale bar = 50 μm. Lower panel, layer depth (µm). One-way ANOVA: EGL, **, *p* = 0.002; PCL, *, *p* = 0.03; IGL, ns, *p* = 0.1. In each layer Bonferroni post-tests were performed and statistical significance for each pair is indicated; (**F**) Purkinje cells immunofluorescence analysis of cerebellar sections using an anti-Calbindin-specific antibody. Nuclei were counterstained with Hoechst. Scale bar 50 μm. Right panel, quantification of PCs density (cells/mm). The analysis was performed in the same animals used in Panel D. One-way ANOVA, **, *p* = 0.002. Bonferroni post-test was performed and statistical significance for each pair is indicated: WT vs. *Ugt1*^−/−^, **, *p* ≤ 0.01; *Ugt1*^−/−^ vs. *Bvra^−/−^Ugt1^−/−^*, *, *p* ≤ 0.05. EGL, external germinal layer; PCL, Purkinje cell layer; IGL, internal granular layer. Data represent the mean ± SD.

**Figure 2 antioxidants-10-02029-f002:**
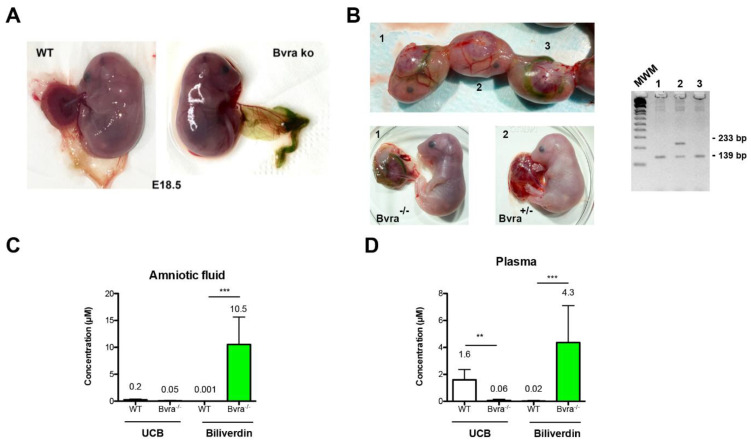
***Bvra* deficiency results in biliverdin accumulation in amniotic fluid and embryos.** (**A**) Biliverdin accumulates in *Bvra^−/−^* embryos at embryonic day 18.5 (E18.5) but not in age-matched WT embryos; (**B**) dated breeding between *Blvra^+/−^* female and *Blvra^−/−^* males was performed. Embryos and placentas were analyzed at embryonic day 18.5 (E18.5). Resulting embryos were *Bvra^+/−^* (embryo 1 and 3) or *Bvra^−/−^* (embryo 2). Only *Bvra^−/−^* embryos accumulated biliverdin in the yolk sac, indicating that biliverdin did not cross the placental barrier effectively. Genotype of the embryos was verified by genomic PCR. (**C**,**D**) Bilirubin and biliverdin levels in amniotic fluid and in plasma. Data represent the mean ± SD. (*n* = 9–11) for each genotype. Statistical analyses were performed by the Student’s *t*-test (**, *p* ≤ 0.01; ***, *p* ≤ 0.001).

**Figure 3 antioxidants-10-02029-f003:**
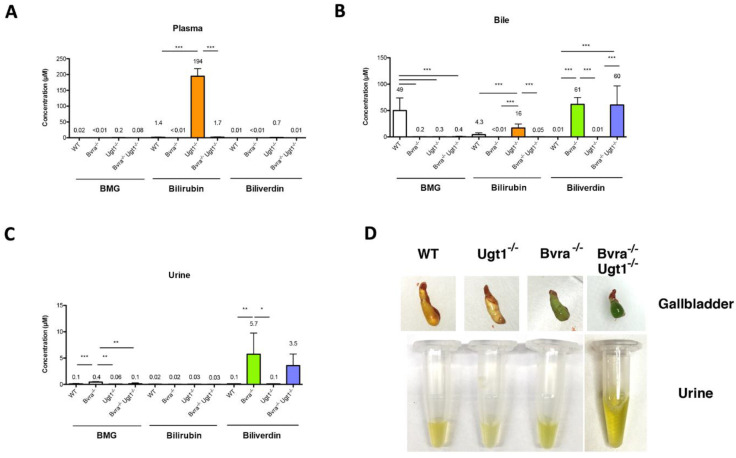
**Bilirubin and biliverdin in plasma, bile, and urine.** Bilirubin monoglucuronide (BMG), bilirubin and biliverdin in plasma (**A**), gall bladder bile (**B**), and urine (**C**) in middle-aged (9-months-old) mice. Data represent the mean ± SD (*n* = 5/gender/genotype; both genders are plotted together). The mean of each group is indicated above each respective bar. One-way ANOVA. Plasma: BMG, *p* ≤ 0.0001, F = 12.59; bilirubin, *p* ≤ 0.0001, F = 622.8; Biliverdin, *p* ≤ 0.0001, F = 86.22. WT vs. *Bvra^−/−^Ugt1^−/−^**,* ns (is indicated for BMG, bilirubin, and biliverdin). Bile: BMG, *p* ≤ 0.0001, F = 43.31; bilirubin, *p* ≤ 0.0001, F = 36.18; Biliverdin, *p* ≤ 0.0001, F = 32.00. WT vs. *Bvra^−/−^Ugt1^−/−^*: *p* ≤ 0.001; ns; *p* ≤ 0.001 BMG, bilirubin, and biliverdin, respectively). Urine: BMG, *p* = 0.006, *p* = 10.10; bilirubin, *p* = 0.6433, F = 0.5662, ns; biliverdin, *p* = 0.0009, F = 9.260. WT vs. *Bvra^−/−^Ugt1^−/−^*, ns (is indicated for BMG, bilirubin, and biliverdin). Bonferroni tests: *, *p* ≤ 0.05; **, *p* ≤ 0.01; ***, *p* ≤ 0.001. (**D**) Representative pictures of gallbladder (upper panels) and urine (lower panels) of WT, *Bvra^−/−^, Ugt1^−/−^* and *Bvra^−/−^Ugt1^−/−^* mice.

**Figure 4 antioxidants-10-02029-f004:**
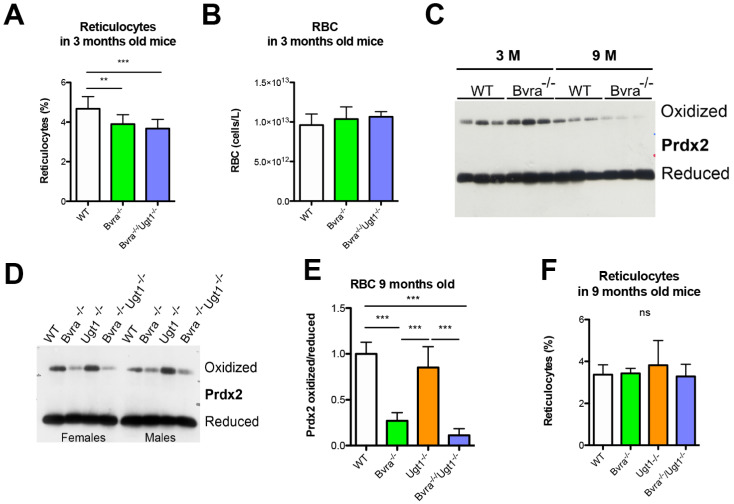
*Bvra* deficiency results in increase in oxidized peroxyredoxin-2 and lower erythrocyte turn over in 3- but not in 9-months-old mice. (**A**) The percentage of reticulocytes, as determined by acridine orange staining of blood in 3-month-old mice. One-way ANOVA, *p* = 0.0006, F = 9.982, Bonferroni post-test, WT vs. *Bvra^−/−^Ugt1^−/−^*, *p* ≤ 0.001; (**B**) RBC counts in 3-month-old mice. One-way ANOVA, *p* = 0.1709, F = 1.887, ns; (**C**) Western blot of oxidized and reduced peroxiredoxin 2 (Prdx2 in erythrocytes of 3- and 9-months-old WT and *Bvra^−/−^* mice. (**D**) Western blot of oxidized and reduced Prdx2 in erythrocytes of 9-months-old WT, *Bvra^−/−^, Ugt1^−/−^* and *Bvra^−/−^Ugt1^−/−^* mice; (**E**) densitometric quantification of the Western blot giving the ratio oxidized/reduced Prdx2. Two-way ANOVA. Interaction genotype-gender ns; genotype, ***, *p* < 0.0001; gender, ns. Bonferroni post-test *Bvra*^−/−^ vs. WT or *Ugt1^−/−^* ***, *p* < 0.0001; *Bvra^−/−^Ugt1^−/−^* vs. WT or *Ugt1^−/−^* ***, *p* < 0.0001; WT vs. *Ugt1^−/−^,* ns; (**F**) percentage of reticulocytes, as determined by acridine orange staining of blood from 9-month-old mice. One-way ANOVA, *p* = 0.4202, F = 0.9692, ns. Bonferroni tests: **, *p* ≤ 0.01; ***, *p* ≤ 0.001.

**Figure 5 antioxidants-10-02029-f005:**
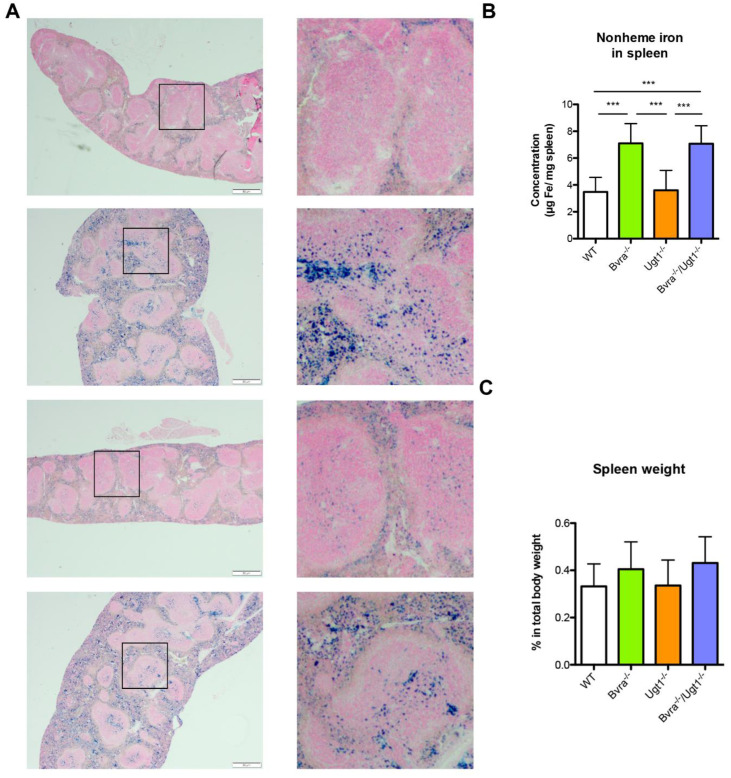
**Iron accumulation in spleen from *Bvra^−/−^* and *Bvra^−/−^Ugt1^−/−^* 9-months-old mice.** (**A**) Representative images of Prussian blue staining revealed iron deposition in the spleen of 9-months-old *Bvra*^−/−^ and *Bvra*^−/−^*Ugt1*^−/−^ mice. The boxed areas are shown at higher magnification in the right hand side of the panel. Scale bar 50 μm; (**B**) quantification of non-heme iron extracted from the spleen. One-way ANOVA, *p* ≤ 0.0001, F = 22.29; Bonferroni tests: ***, *p* ≤ 0.001; WT vs. *Bvra^−/−^Ugt1^−/−^: p* ≤ 0.001; (**C**) spleen weight of 9-month-old mice, normalized by body weight. One-way ANOVA, *p* = 0.1145, F = 2.118, ns.

**Figure 6 antioxidants-10-02029-f006:**
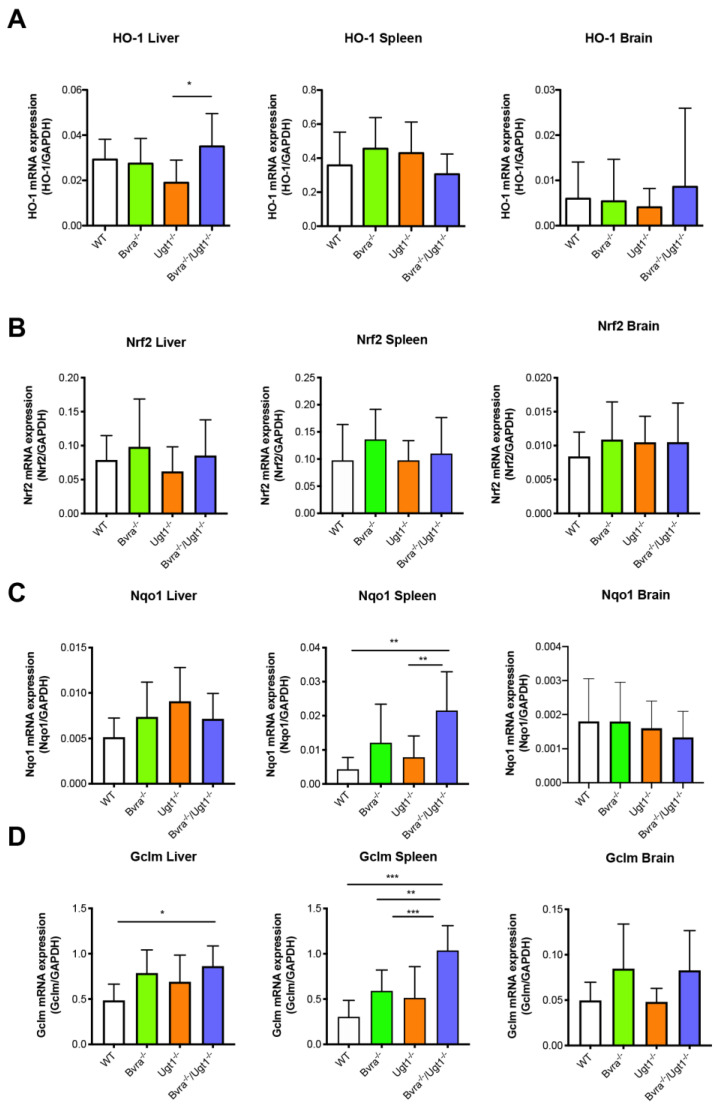
**Gene expression analysis of oxidative stress markers.** (**A**) *Hmox1* (HO-1) mRNA expression levels in liver, spleen, and brain. One-way ANOVA: liver, *p* = 0.0283, F = 3.390, Bonferroni tests: WT vs. *Bvra^−/−^Ugt1^−/−^,* ns; spleen: *p* = 0.1705, F = 1.759, ns; brain, *p* = 0.8114, F = 0.3193, ns.; (**B**) *Nfe2l2* (Nrf2) mRNA expression levels in liver, spleen, and brain. One-way ANOVA, liver: *p* = 0.3650, F = 1.089, NS; spleen, *p* = 0.3167, F = 1.215, NS; brain: *p* = 0.6649, F = 0.5292, NS; (**C**) *Nqo1* mRNA expression levels in liver, spleen, and brain. One-way ANOVA, liver: *p* = 0.1049, F = 2.196, NS; spleen: *p* = 0.0009, F = 6.757, ***, Bonferroni test: WT vs. *Bvra^−/−^Ugt1^−/−^ p* = 0.0013, **; brain: *p* = 0.4100, F = 0.9853, NS; (**D**) *Gclm* mRNA expression levels in liver, spleen, and brain. One-way ANOVA, liver: *p* = 0.0127, F = 4.085, *, Bonferroni test: WT vs. *Bvra^−/−^Ugt1^−/−^ p* = 0.0121, *; spleen: *p* < 0.0001, F = 12.57; Bonferroni tests: WT vs. *Bvra^−/−^Ugt1^−/−^ p* < 0.001, ***, *Bvra^−/−^* vs. *Bvra^−/−^Ugt1^−/−^ p* = 0.0034, **, *Ugt1^−/−^* vs. *Bvra^−/−^Ugt1^−/−^ p* = 0.0003, ***; brain: *p* = 0.0217, F = 3.612, *, Bonferroni tests: NS. Data represent the mean ± SD.

**Figure 7 antioxidants-10-02029-f007:**
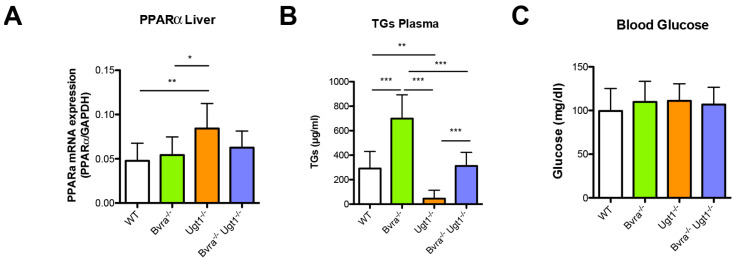
**Differences in lipid metabolism in *Bvra^−/−^* or *Ugt1^−/−^* mice are absent in the *Bvra^−/−^Ugt1^−/−^* double knockout mice.** (**A**) Hepatic PPARα mRNA levels in 9-months-old mice. Data represent the mean ± SD (both genders are plotted together). One-way ANOVA, *p* = 0.005, F = 5.071; Bonferroni tests: *, *p* ≤ 0.05; **, *p* ≤ 0.01; WT vs. *Bvra^−/−^Ugt1^−/−^,* ns; (**B**) triglycerides levels (µg/mL) in serum of 9-months-old animals. One-way ANOVA, *p* ≤ 0.0001, F = 39.19; Bonferroni tests: **, *p* ≤ 0.01; ***, *p* ≤ 0.001; WT vs. *Bvra^−/−^Ugt1^−/−^,* ns; (**C**) glucose levels (mg/dl) in fasted 9-months-old animals. One-way ANOVA, *p* = 0.7094, F = 0.4642, ns.

## Data Availability

The data is contained within the article or Appendix A.

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
