# Peer review of "Long-Term Effects of Biliverdin Reductase a Deficiency in Ugt1−/− Mice: Impact on Redox Status and Metabolism"

_antioxidants, 2021, doi:10.3390/antiox10122029_

Round 1

Reviewer 1 Report

The manuscript entitled:“ Long-term effects of biliverdin reductase A deficiency in UDP-glucuronosyl transferase knockout mice: impact on redox status and metabolism”, submitted by Bortolussi,G. et al., aimed at understanding the systemic consequences of deleting biliverdin reductase (BVR) in an experimental mouse model.

Authors performed a well-designed study, as not only short-term effects resulting from BVR-deletion, but also long-term effects were analyzed, in order to answer the principal question, whether targeting of BVR would be a suitable approach to manage neonatal jaundice. Another smart approach is the use of mice with a deficiency in UDP-glucuronosyl transferase (UGT), allowing to understand the role of elevated bilirubin (BR). The authors present convincing data regarding the functionality of their experimental model system. UGT/BVR double knock out mice displayed severe unconjugated BR levels during the late state of the fetal development, which were fatal when remaining untreated. However, regarding the long-term effects of BVR deletion, contrary to what one might have expected, neither oxidative stress levels, nor metabolic status were altered in BVR-KO mice indicating safety of BVR inhibition.

The manuscript is well written, data are presented comprehensively and conclusions are supported by convincing data including those displayed in the extensive supplementary material. However, I do not completely share the conclusion of the authors that long-term BVRA inhibition is safe. Although the data of this elegant study definitively show that mice do not require BVR to maintain metabolism and redox homeostasis under physiological conditions, authors cannot exclude a role of BVR under pathological conditions. Therefore, two aspects need to be discussed in more detail: 1) mice lacking BVR activity display splenomegaly and strongly increased levels of non-heme iron in spleen. This indicates that BVR may be associated with iron mobilization and export, similarly to what has been shown for HO-1. Here it would be interesting to study splenic expression levels of iron exporter ferroportin as well as hepatic expression of hepcidin. Irrespectively, splenic iron overload indicates a potential risk factor, which authors should not ignore. In this respect it would be advisable to study oxidative stress markers in the spleen and the changes of other antioxidant systems, such as glutathione ect in erythrocytes and spleen. 2) Further, authors do not consider that serum levels of BR strongly differ among species. Healthy mice display an about 10-fold lower BR content compared to humans. Contrary to rodent species, an evolutionary advantage of increased BR levels is obvious for humans. This point indicates a different role for BVR in these species. Despite of being an excellent model system, mice appear to be not entirely suitable to study the role of BR and BVR. Therefore, the findings of this study, which was done in mice, do not exclude that BVR may play a more relevant role in humans. Additionally, a beneficial role of elevated BR has been shown in pathologies that are associated with increased oxidative stress, altered erythrocyte half-life and increased inflammation. Authors did not analyze the role of BVR deletion in a comparable pathological context. Therefore, authors should take into account these aspects in more detail in the discussion section.

Minor points:

  • Please provide images of the histologic analyses with higher resolution.
  • Please indicate the number of animals investigated for each parameter in the legends to the figures (as was done for Fig. 1 A and C).
  • I am not a friend of indicating non-significant differences. To my opinion, these indicators should be removed.
  • Line 271: please exchange the letter ‘a’ for an ‘e’ after “eliminate”.
  • Line 480: please check the formatting. It appears that a part of a sentence has been lost.

Author Response

Response to reviewer 1:

We thank the reviewer for his/her time and effort and kind words.

The reviewer does request two aspects to be discussed in more detail.

1: The splenomegaly and the increased non heme iron accumulation seen in the spleen of Blvra deficient mice.

We do agree with the reviewer that Bvra deficiency results in a significant increase in iron accumulation in the spleen (Figure 5) suggesting Bvra may have a role in iron mobilization and export. Our initial ideas on the cause of this were similar to those mentioned by the reviewer that there would be a problem with iron mobilization. However, in contrast to HO-1 deficient mice that accumulate iron in liver and kidney but not in the spleen (Poss et al PNAS 1997), in our Bvra-/- and Ugt1-/- Bvra-/- mice we only see an increase of iron in the spleen, with levels of HO-1 mRNA similar to WT in both spleen and liver (Figure S7). As proposed by the reviewer we have determined the expression of hepcidin mRNA in the liver. The qRT-PCR data showed the highest mean hepcidin mRNA level was present in the liver of the WT animals. Since increased hepcidin levels results in reduced levels of ferroportin, the iron exporter, in the spleen, these data do not provide an explanation. Since presence of non-heme iron in the spleen is normal, the most likely explanation for the higher levels in the Bvra-/- mice would be an increased turnover of heme in these cells suggesting an increased erythrocyte turnover, that would be in agreement with the lower presence of oxidized Prdx2 in Bvra deficient mice.

We do agree with the reviewer this iron accumulation needs to be considered when inhibiting Bvra and have added the Hepcidin data to the Results Section.

Page 15, lines 382-385: “The mRNA level of Hepcidin in the liver, inhibiting intestinal iron uptake and iron export from the spleen, did not differ significantly, suggesting the absence of a major change in iron metabolism (Figure S7F) (38).“

Page 22, lines 617-620: “The splenic iron accumulation observed in 9-month old Bvra and Bvra-/-Ugt1-/- did not result in increased inflammation but since the mechanism could not be clarified, monitoring serum iron content and ferritin should be considered during long-term Bvra inhibition.”

Regarding the weight of the spleen in the different strains of mice, as shown in Figure 5C, there was a minor increase in the weight of the spleen (normalized by body weight) in both Bvra KO strains (n=10-11 mice per group), but this difference did not reach statistical significance (Spleen weight of 9-month old mice, normalized by body weight. 1-way Anova, P=0.1145, F=2.118, ns).

To improve the clarity of the message, we modified Figure 5A by including more representative sections of the spleen. We have also included a close up of the spleen showing the iron accumulation in more detail.

2: the authors do not consider the large difference in serum levels between species. An evolutionary advantage of increase BR levels is obvious for man.

The serum bilirubin level in rodents is much lower than the normal level in man. This may indeed suggest a more important role of bilirubin as an antioxidant in man. However, birds lack BVRA and some species, like some parrots, do reach a similar age as humans. In any case, the scope of this study was to investigate if inhibition of Biliverdin reductase in the context of severe hyperbilirubinemia could be a safe and feasible treatment option. The potential adverse effects of BVRA deficiency or of BVRA inhibition in the context of other (inflammatory) disease was not investigated also because we only expect this treatment to be applied for hyperbilirubinemia. Thus, the double knock-out mice is the most relevant model in this study and does show the feasibility of this approach. In these DKO mice the serum bilirubin levels were comparable to those in the WT mice. Also the two BVRA deficient subjects reported in Greenland had normal or even somewhat increased serum bilirubin levels (albeit that that study did not determine if this was unconjugated or conjugated bilirubin). This indicates that even in complete absence of BVRA, some bilirubin is formed. Further research will be needed to resolve the mechanism and to learn if this amount is sufficient to be beneficial in the pathologies mentioned by the  reviewer.

We do agree that these results should be taken with some caution and we have added this in our conclusion at the end of the discussion.

Page 22, lines 614-617: “The transient mild oxidative stress in adult Bvra deficient mice suggests long term inhibition of this enzyme to reduce serum bilirubin levels seems feasible albeit that the low bilirubin levels compared to man may imply that its role as an anti-oxidant may be less important in mice”.

Minor points:

We have added images with a higher resolution, Figure 5A.

Since the number of animals was not the same in all cases, to avoid rendering the Figures and Legends too heavy, we included these data as a new Supplementary Table (Table S3).

At line 271 “elimita” has been corrected to “eliminate”. We thank the reviewer for noticing this.

Line 480: There was indeed a part missing due to the conversion into pdf. We have corrected this and thank the reviewer for noticing this.

Reviewer 2 Report

The results of body weight in females are very interesting. A figure should be added with the bilirubin levels of the females, as well as the estradiol levels in the blood. Above all, taking into account that high levels of estradiol, in addition to increasing biogenesis, as discussed in line 428, is also capable of regulating antioxidant levels and thus modulate oxidative stress, since estradiol per se can act as an antioxidant.

Author Response

Reviewer 2.

We thank the reviewer for his/her time and help to improve our manuscript.

Based on the Table of the Review Report Form, the reviewer indicates that information on the background and some references in the Introduction are missing. We do apologize for that but since it is not stated which information is lacking it is difficult to improve the manuscript. The main goal of this study is to investigate if Bvra inhibition would be a feasible treatment for severe hyperbilirubinemia, by blocking bilirubin production. We have tried to include all information focussing on this scope. We have included new references and modified the text in the Introduction Section (Page 4, Lines 102-106). Although we are aware that UGT1 has a major role in metabolising many compounds, in view of our aim, we have limited the introduction to bilirubin metabolism and the role of Bvra. We are prepared to include additional information that the reviewer may find essential in view of this scope and we did include an additional paper showing presence of asymptomatic Bvra deficient subjects in the Inuit population (New Reference 21).

The reviewer asked us to include a figure showing the serum bilirubin levels in the female animals in combination with the estradiol serum levels. Upon splitting the serum bilirubin levels in male and female Ugt1-/- mice we do not find any difference (see Figure added below). Therefore, we think showing the mean for all animals will be sufficient. We have added a sentence stating this in the manuscript. The difference in weight of Ugt1 deficient females was unexpected. We have not determined the estradiol levels in serum of female mice and due to all analysis done on these samples no residual material is available to do this. As mentioned, the aim of this study is investigating if Bvra inhibition could be a feasible approach to treat Ugt1 deficiency. The most relevant genotype is the double knock out. Since there is no significant difference between the Ugt1 -/- and the DKO in body weight this observation is somewhat outside of the scope of this study. We do however agree that it is an interesting observation for a follow up study.  

Round 2

Reviewer 2 Report

The authors have not addressed or justified the absence of the new determinations recommended in the previous review. Furthermore, the authors have not provided the new version of the supplement. Below I return to the recommendations that the authors should address to improve the quality of the article. I outlined below some of my previous comments that should be addressed.

-Please, review the abbreviations, and eliminate the unnecessary ones (HIC, LIC and MIC among others).

- In the statistics section, provide the test carried out for the normal distribution prior to the ANOVA.

-The differences in the blood hematological analysis (figure S4) between 3 and 9 months should be discussed in depth.

- The oxidation of prx2 is not a sufficient measure to assess the redox status, especially if the intention is to put it in the title of the article. Some examples would be: NAD+/NADH and NADP+/NADPH redox ratios, GSH/GSSG, redox-sensitive green fluorescence proteins OR fluorescein-5 maleimide. Alternatively, antioxidant enzyme activities such as the Thioredoxin system, among other options, could be measured.

-NRF2 is a so-called oxidative stress sensitive protein. When there is stress, it is able to get rid of keap1 and go to the nucleus to transcribe antioxidant genes, but also others genes such as Ho-1. It would be very interesting to measure the activation of NRF2 (appearance in the nucleus by wb) or, failing that, levels of gene expression of Nfe2i2 to relate the increase in Ho-1 of the liver with Fgf21 (an inducer of browning in white adipose tissue) and the results of "oxidative stress" shown by prdx2.

-Supplemental Table 2 presents the misspelled gene abbreviations. Please, correct the names of the genes of that table and also from the other figures of the article following the appropriate nomenclature (mouse MGI).

-To be able to put in the title the impact of the absence of biliverdin reductase A in Ugt1-/- mice in the metabolism, some other parameter of metabolic character should be analyzed. Among those that stand out Oxygen consumption (metabolic cage), body weight during the 9 months, weight of some tissues such as the liver, spleen ..., OGTT / ITT…

-In line 572 is state that the TG liver were in the normal range, but there is no figure showing that data.

Round 3

Reviewer 2 Report

General comments

I reviewed the manuscript three times. I can see, that authors really improved their paper, therefore in current version I accept this manuscript.